# Performance evaluation of automotive product development team members based on a PLM system: A case study of M Automotive Products Company

**Debao Dai, Shihao Wang, Yinxia Ma**◯*

Faculty of Management, Shanghai University, Shanghai, China

* yinxia.ma@foxmail.com

**Data Availability Statement:** All relevant data are within the manuscript and its Supporting information files.

## Abstract

Based on product lifecycle management (PLM) theory and social comparison theory, this paper constructs a performance ranking model of automobile product development project team members, uses an active server provider (ASP) to develop the system, realizes the online operation and real-time analysis of the performance management system, and solves the problems of the low efficiency, low morale and unfair assessment of product development team members caused by traditional performance assessment. The performance management platform for the team members of the automotive product development project uses a PLM system to realize a systematic and standardized list of project functions and provide a performance appraisal management system that can be evaluated and fairly compared for project team members. Based on the performance appraisal practice of automotive product development project team members of M Automotive Products Company, this study verified the feasibility of the ranking model's transformation to work habits based on the predetermined and result data of the mission completion rate, the punctuality rate and the degree of improvement of the management level of product development project team. Through the tracking of the team performance ranking function, it is found that the model can solve the problems of untimely assessment and insufficient incentives in the current traditional performance appraisal, enhance the overall enthusiasm of the team, and give full play to the subjective initiative of the team.

## 1. Introduction

Project management methods, project team performance evaluation, project team performance incentives, and PLM system management are relatively independent in terms of their explanations and interpretations. However, with the gradual improvement of the modern supply chain development system, the continuous improvement of the industrialization level of China, and the gradual improvement of product development capability, automobile development projects are no longer carried out by an original equipment manufacturer (OEM) alone

**Funding:** This study was supported by the General Program of the MOE Layout Foundation of Humanities and Social Sciences: An Empirical Study on the Divergence and Convergence of Dynamic Process of Online Learning in the form of a grant awarded to DD (17YJA880014).

**Competing interests:** The authors have declared that no competing interests exist.

[1]. In the fiercely competitive automobile market in China, the annual number of new automobile models has reached more than 70 since 2005. As the main management leaders in the industry, to ensure the competitiveness of products, new model development projects exist, which have short durations, high development costs, and multiproject synchronous development, leading to high requirements for the project management of automobile product development [2]. Especially in the development of new products, the automobile industry has become the project management benchmark of all new product development projects, and with the rapid development of the supply chain, the traditional organizational structure and project management method cannot support the operation of enormous automotive development projects. Therefore, the automobile manufacturing industry needs to realize product lifecycle management, and the members of the product development project team are the main performers of product lifecycle quality assurance, so it is very important to evaluate the performance of team members [3]. In the current management process, product development management assessment delays and incentives that are not timely have become one of the most important problems in today's society. The traditional product development team performance evaluation model mainly evaluates the economic benefit of product development output, customer satisfaction, internal research and development cost, team time management and knowledge product value, and has the following problems [4–7]:

(1) Slow evaluation method. Team members can only be evaluated when the project is nearing completion.

(2) After the completion of the project, the value of time management and knowledge products is difficult to be traced back and correctly evaluated.

(3) As time goes by, the overall value of the project will be iterated by new products, and the value will be weakened. Product development projects are characterized by a long lifecycle and great difficulty, which directly affect the enthusiasm of development team members. The unfairness of performance appraisals will reduce the satisfaction of knowledge employees; lead to an increase in enterprise talent turnover; bring about a series of problems, such as the reduced efficiency of project teams, repeated work and new people taking over; and lead to great management losses. The lag of traditional performance appraisal and the characteristics of long and difficult life cycle of product development project directly affect the enthusiasm of development team members [8–10]. The unfairness of performance appraisal will reduce the satisfaction of knowledge workers; Lead to the increase of enterprise brain drain; It brings a series of problems such as reduced efficiency of the project team, repeated work and new people taking over. And resulted in huge management losses [11–13]. Therefore, in the face of this problem, the research needs to start from the following two points:

(1) How to realize the timeliness of performance appraisal

(2) How to ensure the fairness of incentives

In the existing literature, it is found that job stress can influence employees from two aspects: professional motivation and professional burnout [14, 15]. Product research and development team members are relatively highly educated teams. It can be concluded from the literature that workplace pressure for knowledge workers tends to push them forward in the form of motivation for product research and development team. An equation can be constructed: creative pressure = motivation for product research and development team. As knowledge-based and high-quality employees, development team members need to ensure their stability. In terms of performance evaluation and incentives, they also need to provide a

special assessment mechanism according to local conditions. Development team members can generate incentive effects under pressure, so a method needs to be developed to meet the requirements of fairness, timeliness and pressure mechanisms at the same time. To solve these problems, this study constructed a set of mature project management systems to ensure the fairness of assessment and incentives of timeliness. This management system provides team members with standard work tasks and work requirements and uses a PLM system to standardize the assessment and provide overall transparency so that team members can better understand their own progress and situation and reflect on the project situation and team progress in a timely manner. Through PLM system, in the current digital and intelligent environment, improve the efficiency of project management, performance management and performance incentive, help the project manager to manage the team more quickly and efficiently, evaluate and motivate the team. In addition, this paper uses project management to transform the current digital real-time management into employee performance evaluation tool to fill the blank of social comparison theory application in project management process.

## 2. Literature review

A product lifecycle management system is an information system that performs a complete set of product management processes, from product development to production implementation and change management [16]. First, as the key product data input of enterprise resource planning (ERP), supplier chain management (SCM) was established to ensure the consistency of upstream and downstream data, and project management (PM) has been used to ensure the timely maintenance of engineering data. Problems such as inconsistent upstream and downstream information, opaque project schedules and actual situations in the current traditional product development management process have been solved [17–19]. These problems have been improved one by one by digitization and informatization to address the current problems of multilocation production and collaborative development faced by enterprises. Foreign scholars have also made positive comments about the development management of enterprises through the use of PLM systems, promoting product data management and improving development process management [20]. In addition, some studies have used standardized and digital operations of project management to assess development staff and provide timely and rapid feedback on the key performance indicators, the key tasks and the project maturity [21]. This has a profound impact on the intelligent manufacturing of enterprises, as it not only provides enterprises with the feasibility of business process optimization with the process being the core and the environment for collaborative project management but also provides no guarantee for the creation of a knowledge base, management and the sharing of knowledge to better enhance the overall management competitiveness of enterprises [22, 23]. A PLM system can reflect the work task, project plan and completion situation in real time and transparently, and employee performance assessment can also be made clear through performance, attitude and ability assessments. However, the method of employee performance evaluation through a PLM system is not integrated, and the inconsistency between data and reality further intensifies the contradictions in current management research. Therefore, this study will establish a set of performance evaluation mechanisms [24–26], set up performance incentive means [27, 28] based on the characteristics of the product development team, build a set of engineer performance ranking mechanisms [29] using social comparison theory, and conduct system customization development based on the PLM system of M Automotive Product Company.

Development team members, as the core creative factors in the project team, have become the core resources preferred for technological innovation. Development staff members, as highly educated and creative talent, have enthusiasm and creativity that act as the key factors

in the success of product development management activities [30, 31]. Therefore, the performance evaluation of development personnel is the factor that can most influence the success of a project. Some studies suggest that individual and team motivation for development team members can maintain the vitality of enterprise development and innovation, which also indicates the important role of team performance evaluation and salary incentives in the motivation of development staff [32–34]. Team performance incentives can better promote team members to work together and try to target important factors and establish a development personnel performance appraisal scale, which can realize the performance, ability, and attitude of performance incentive development staff to ensure the fairness of salary incentives. However, based on full consideration of the positioning of development personnel as the employees with the most knowledge in enterprises, the characteristics of knowledge-based employees can be further summarized as follows: they have greater potential value, higher personal quality and life pursuit, stronger willingness to flow, and distinct creativity and autonomy, and it is difficult to directly measure and evaluate their labor achievements [35]. In addition, high-quality employees not only want to make money but also hope to achieve personal development and improvement in their work and are willing to challenge unknown fields. In terms of performance incentives, they should pay attention to the following five aspects: work incentives, organizational incentives, salary incentives, cultural incentives, and learning and knowledge sharing [36–41]. To improve the performance of development personnel, they should transform from being managed, followed by self-managing and then to managing the team.

Based on the above analysis, this study will use cognitive evaluation theory [42, 43] and fairness theory [44, 45] to adjust the working habits of the project team and realize the timeliness and fairness of performance evaluation through the PLM system. Then, a data tracking survey is conducted from functional categories to provide the most intuitive and objective results in a data-oriented way. Finally, under the social comparison theory model [46, 47], referring to the literature on the organizational fairness cultural path, the driving force and changes in the habits of team members were analyzed, providing a reference for business promotion.

## 3. Method

PLM system combined with cognitive evaluation theory, equity theory, social comparison theory, product research and development team members performance evaluation model was constructed to researchers the day-to-day work of the data into big data analysis, combining with the implementation of enterprise itself has the information function, achieve the goal of enterprise management digital integration, will manage the accuracy from the project direction to personnel management direction, And improve the responsibilities of administrative leadership, improve the efficiency of project management [48–52].

### 3.1 Project management requires input

Based on social comparison theory and the requirements of project management, this study will establish a performance evaluation model to achieve the quantitative evaluation of project team ability and project members' performance level. To ensure that the project can be completed on time and with high quality, time, cost and quality must be appropriately managed. The steps involved in this process are as follows:

(1) Project management process method: collect the main indicators and key tasks of the project, and clear the basic process and output requirements for project management. As the most critical part of performance appraisal, the project management process belongs to the

basic logic and core of enterprise management. Therefore, it is of great importance to clarify the project management process to achieve accurate followup appraisal results.

(2) Relationship between key project tasks and project planning: the list of key tasks used for project maturity management in project management and the person in charge of the task shall be defined according to project management requirements. Key tasks undertake the risk management of the project and serve as an important factor in assessing the competence of project members.

(3) Define the project role list according to the organizational structure: define the role list of project team members according to the organizational structure to prepare to define the target work content according to the function. The project role list is used to categorize functions and assign responsibility for task management.

(4) Standard task output template: output the task list according to project management requirements, and assign to the assigned project roles according to the management requirements to ensure project quality output.

## 3.2 Team performance model construction

As shown in Fig 1, during the establishment of this model, the project cost and the value assessment of the project itself are not considered at first. According to the enterprise management situation, priority should be given to the improvement of the overall performance and

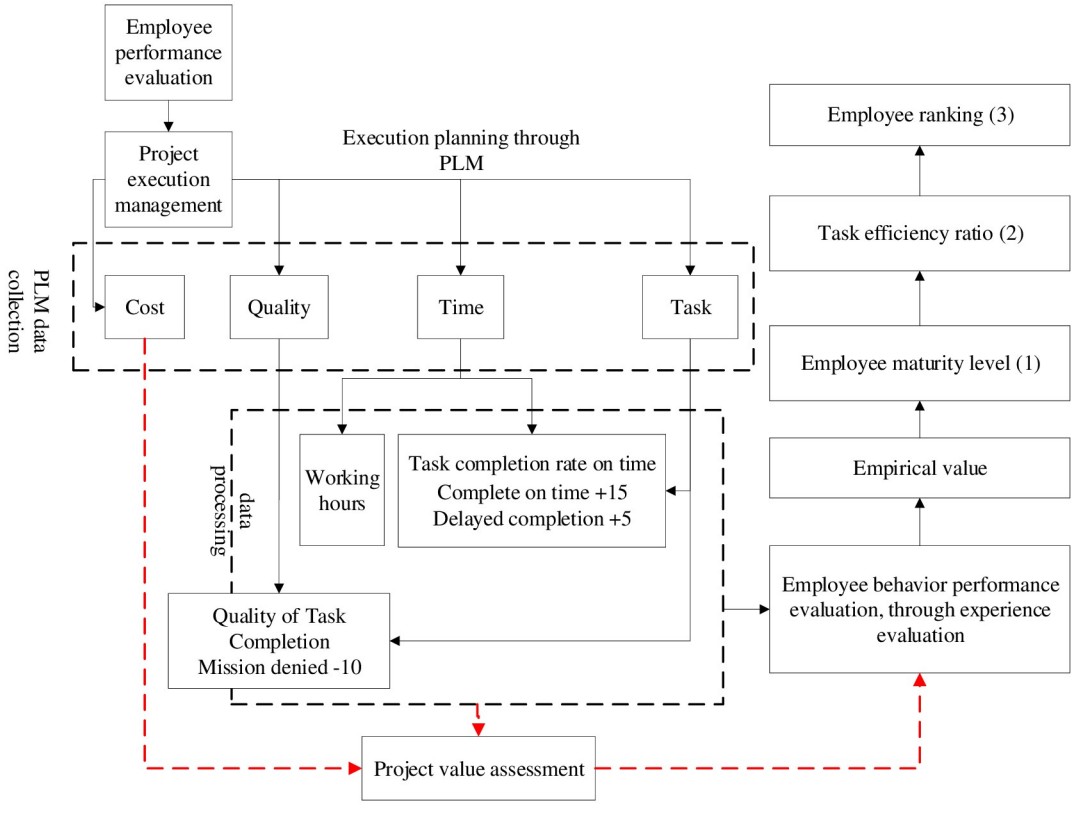

**Fig 1. Research model.**

ability of project team members, and then, project cost should be gradually introduced to evaluate the project team. In this model, tasks are automatically assigned according to process requirements, and time is input according to the time required for project management. Then, according to these time requirements, the on-time completion rate of team members is scored. A ladder scoring method is created; that is, 15 points are obtained if the task is completed on time, and only 5 points are obtained if the task is completed late. In addition, to better assess the quality of task output, the superior manager needs to approve the task. When the task is returned in the approval process, the system automatically deducts 10 points. The rate of return of the task can evaluate the working ability of team members and employees' own business ability according to their total points. Here, (1), (2), and (3) are as follows:

(1) Due to the different number of tasks assigned by different departments in the project, grade upgrading is defined according to different departments, which are divided into 5 levels.

(2) Actual score/theoretical score: level 1 score $^*$70%, and level 2 score $^*$85%. According to efficiency rather than to performance rating, A: > 85%, B: > 75%, C: > 65%, D: > 55%, and E: < = 55%.

(3) Sorting score: performance base score + actual score/10000+(grade $^*$ efficiency ratio)/10 (A: 2, B:1.5, C:1, D:0.5, E:0).

## 3.3 Staff rating system

Combined with the level of the enterprise itself, an integral upgrade system is established through the hierarchy to quickly identify an employee's ability level. Based on the actual research objects, this study defines a five-level system in the employee hierarchy according to the five-level engineer rating rules of the enterprise. In accordance with the management requirements of the human resources department, the level distribution method is used to allocate the proportion of the number of employees in the hierarchy, as shown in Fig 2.

Eq 1, for calculating the total score of employees, is as follows:

$$\text{Total score} = \sum \text{Number of delayed tasks} * 5 + \sum \text{Number of on} - \text{time tasks} * 15$$
$$- \sum \text{Number of rejected tasks} * 10 \tag{1}$$

## 3.4 Performance evaluation analysis

Team members obtain credits according to the number of tasks they have completed. To better promote the overall completion of engineers, the work attitude of employees will be assessed,

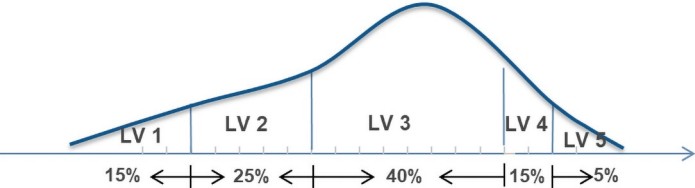

**Fig 2. Proportional distribution of the number of rank personnel.**

and the task efficiency ratio will be calculated. The calculation formula is shown in Eq 2:

Task efficiency ratio

$$= \frac{\sum \text{Number of delayed tasks} * 5 + \sum \text{Number of on} - \text{time tasks} * 15 - \sum \text{Number of rejected tasks} \times 10}{\sum \text{Total number of tasks} * 15} \quad (2)$$

After comprehensively considering that employees at a low level (low work capability maturity) need support from other colleagues in their daily work and combined with the requirements of the personnel department in terms of employee performance appraisal objectives, the task efficiency ratio of low-level employees will be dealt with in the following ways, as shown in Eqs 3 and 4:

$$\text{Level 1}: \quad \text{Task efficiency ratio} = \frac{\text{Total score}}{\sum \text{Total number of tasks} * 15} * 70\% \quad (3)$$

$$\text{Level 2}: \quad \text{Task efficiency ratio} = \frac{\text{Total score}}{\sum \text{Total number of tasks} * 15} * 85\% \quad (4)$$

According to the final task efficiency, to evaluate the performance of A: > 85%, B: > 75%, C: > 65%, D: > 55%, E: < = 55%

### 3.5 Building a ranking model

In the overall ranking model, the task efficiency ratio should not be the only factor considered, and thus, the overall evaluation of the comprehensive ability of employees should be carried out. With the goal of improving the on-time rate of engineers' tasks, taking the work performance rating as the basis and considering the daily load of employees and the differences in their scores, the emphasis of ranking score calculation is divided into three aspects:

(1) Performance level: the task on-time rate of the overall response engineer.

(2) Total score: The total score comes from the number of tasks and is ranked by load at the same level.

(3) Staff level: weighted by level and ranked by task maturity under load.

In Eq 5, the ranking score is set, in which the basic performance grades are A = 2, B = 1.5, C = 1, D = 0.5, and E = 0.

$$\text{Ranking score} = \text{Performance level base score} + \text{Total score}/10000 + \text{Staff level} \\ \times \text{Task efficiency ratio}/10 \quad (5)$$

## 4. A case study

### 4.1 Status analysis of M Automotive Products Company

M Automotive Products Company is a joint venture established by Shanghai Automotive Industry Corporation (SAIC) and Germany's M Group, which integrates the research, development and manufacturing of automotive heat exchange products into an auto parts enterprise. Since the joint venture in 2004, the company has averaged more than 160 new product development projects every year, serving more than 30 vehicle customers, including Volkswagen, General Motors (GM), Bayerische Motoren Worke (BMW), Mercedes-Benz, Jaguar, Land Rover, and Volvo. With Shanghai as the development center, the company has covered

10 domestic production areas. The company has more than 2000 people who provide continuous technical and management support for the enterprise to maintain steady growth in the industry.

M Automotive Products Company uses the German global project management method of a product development process (PDP). Through this process, it manages the whole lifecycle of products from development to after-sales maintenance. The company is praised by the industry for its excellent management ability and management style. However, with the continuous increase in the number of projects, as well as the expansion of the industry in recent years, project management has become increasingly difficult, and multicoordination and version control have become unavoidable problems. In this context, in 2016, enterprises started to develop the PDP-PLM platform and officially put it into use in 2017, but there are still many problems:

(1) Traditional work methods are deeply rooted. In the process of systematic management, employees still use management methods outside the system to conduct their daily work.

(2) The enterprise uses a balanced matrix project team structure. In the process of project management, it is difficult for the project manager to break away from the administrative management path to effectively manage and control the project.

(3) There is no project salary incentive for the product development team, and management functions are assigned to their respective functional departments, which increases the difficulty of project management.

## 4.2 Implementation of the evaluation system for team member performance

**4.2.1 System implementation.** The performance evaluation system was developed from April to June, during which time it was gradually generated as follows:

**User information list:** the list of active users in the current system is generated, and user interface permissions are established through the relationship between superior and subordinate employees. Engineers can see their own information, administrative directors can see team information, department managers can see department members' information, and ministers can view the whole department.

**Project information assignment:** the matching of engineer and project information is completed, and the project information and project number of the project in which the engineer is involved is obtained through a single interface.

**Time plan allocation:** according to the project information, the time plan for project planning can be reflected through the project list, and then, the corresponding task list of the project can be obtained.

**Comparison and analysis of task completion:** the completion times for more than 18,000 completed tasks in the system are obtained and compared with the planned times.

This paper uses Python to obtain the task completion rate and on-time rate information and to return it to the user. According to the task data and the design requirements of the model, the task completion rate and return information are calculated as a whole, and then, the job functions in the system are classified and sorted to finally obtain initial calibration data.

**4.2.2 Case implementation.** Formulation of the assessment method: according to the requirements of enterprise management, the project management process and key tasks are defined, the project role of the team is determined, and the task output template is defined. After discussion and confirmation with the management department, the assessment method is determined through the review and finalization of enterprise management.

**Full staff training:** according to the management requirements of the assessment method, the training of all project members ensures the transparency, standardization, and fairness of the assessment. All employees find their position according to the requirements and start the system assessment.

**Introduction of performance salary:** after being confirmed by the human resources department, the performance salary reward system of the project team is introduced to encourage the first, second and third most productive staff members at the material level. In addition, a medal mechanism is used for monthly and annual competitions regarding employees' ability, and management awards these medals as a spiritual incentive.

**Introduction of management requirements:** management at all levels of the enterprise should be able to instill the idea of a ranking mechanism. According to the data generated by the system, managers should be able to understand, promote, and finally import a ranking mechanism into the annual performance of employees based to conduct ranking tracking.

**PLM system implementation:** the planned functions in the system, including the full staff leaderboard, personal situation board, etc., are implemented Employees can see their performance level, task efficiency ratio, total number of tasks, annual ranking, personal ability level and other information in the system.

**4.2.3 Initial calibration.** As of December 2019, employees participating in the assessment included the following: the development product design team of 64 people, the development process planning team of 38 people, the business management team of 25 people, the procurement team of 14 people, the quality management team of 59 people, and the logistics team of 31 people. According to the proportional allocation requirements of the number of personnel at each level and the corresponding functional scores of each role in the system, the grade points of the main project team members are divided, as shown in Table 1.

According to the corresponding roles, the assessment of the initial level and performance ability of project team members and the role performance ratio of the first version were completed in June 2019. The results are shown in Fig 3.

According to the post functions, the post task closure rate and task efficiency ratio of the first version are obtained, as shown in Figs 4 and 5.

## 4.3 Team members' performance evaluation results output

**4.3.1 Employee performance level analysis.** The team performance evaluation system was launched from June to December 2019, and the overall performance level greatly

**Table 1. Post functional level division.**

| Group | Level 1 | Level 2 | Level 3 | Level 4 | Level 5 |
|---|---|---|---|---|---|
| Business | 0 | 100 | 200 | 400 | 700 |
| Logistics | 0 | 100 | 200 | 700 | 2000 |
| Process planning | 0 | 100 | 200 | 600 | 1400 |
| Product design | 0 | 100 | 200 | 400 | 700 |
| Procurement | 0 | 100 | 200 | 700 | 1400 |
| Quality | 0 | 100 | 200 | 600 | 1400 |

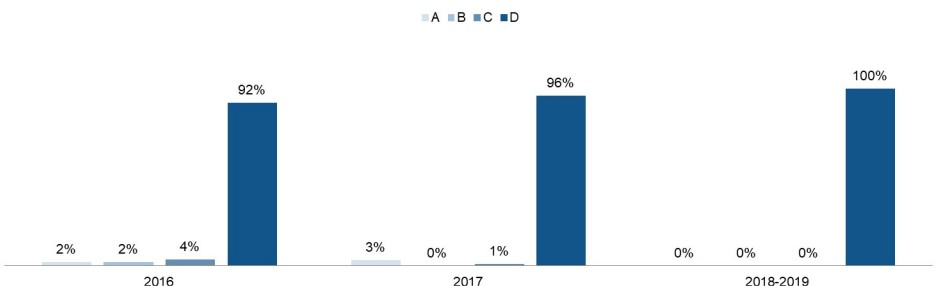

**Fig 3. Role performance ratio (2019.6).**

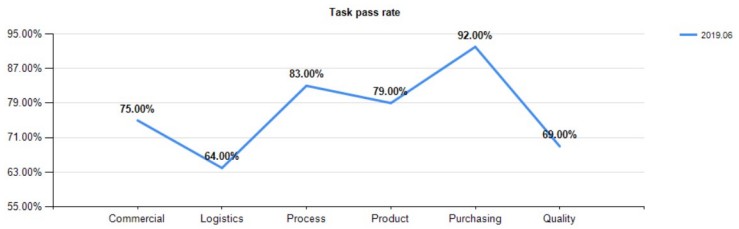

**Fig 4. Closure rate of post functional tasks (2019.6).**

progressed, as shown in Fig 6. The percentage of A-level performers increased by 18%, B-level performers increased by 21%, C-level performers increased by 23% and D-level performers decreased by 62%. In terms of the proportion of qualified employees, their performance significantly improved.

**4.3.2 Proportional distribution of employee rank.** In the process of advancement, an increasing number of project members and project managers volunteer to join the performance evaluation system for ranking competition. In addition, new employees directly join the performance evaluation system according to the new requirements, leading to an increase in the proportion of level-1 employees. Compared with level 4 and level 5, level 3 shows a significant increase in the proportion of employees and has become the backbone of the team. Among employees, the proportion of level-5 engineers increased by 9%. The proportion of the number of employees at different levels is shown in Fig 7.

This paper analyzed the product development team, the development team and the business management team from the perspective of all levels of personnel distribution. Three teams

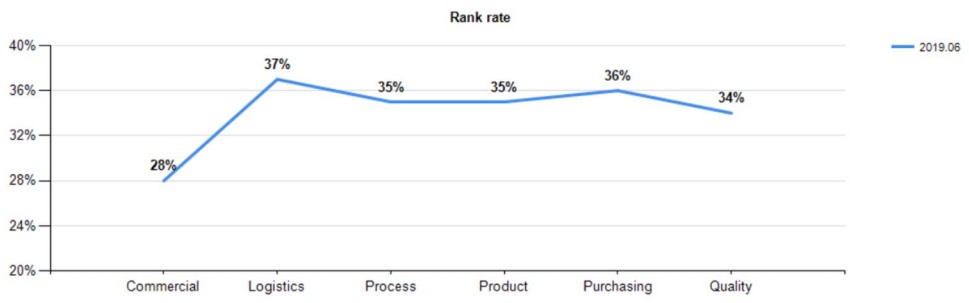

**Fig 5. Efficiency ratio of post functions to tasks (2019.6).**

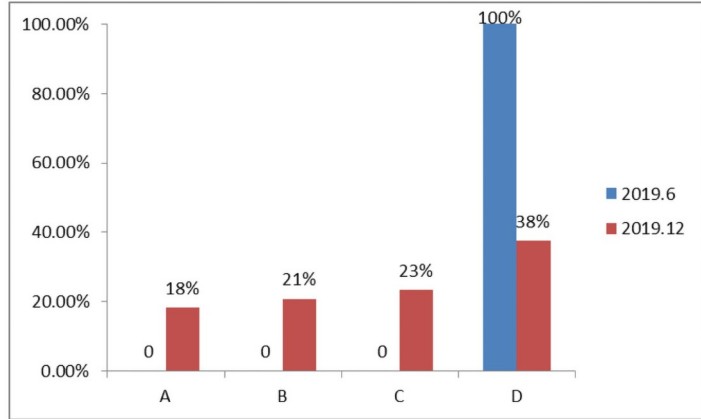

**Fig 6. Ratio comparison of role performance (2019.12).**

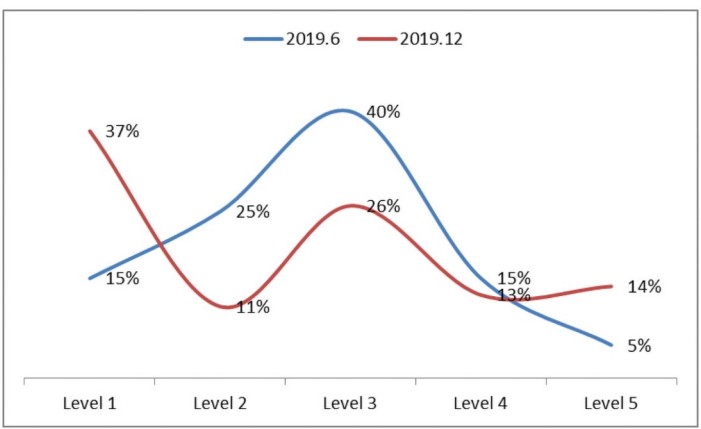

**Fig 7. Ratio of the number of ranks (2019.12).**

revealed a general trend of moving toward level 5, and this general trend led to an increase of 14% in level-5 employees, an increase of 11% in development process employees, and an increase of 11% in business management employees, as shown in Fig 8.

**4.3.3 Task closure rate change analysis.** In the evaluation process, it can be seen from the monthly task closure rate chart that with the use of a scoring mechanism and hierarchy, self-cognition theory gradually has an impact on users' work in daily life. Users gradually formed

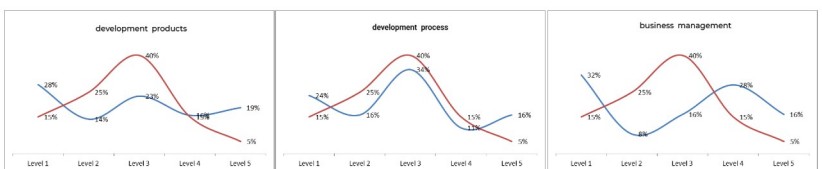

**Fig 8. The proportion of development products, development process, and business management personnel grades.**

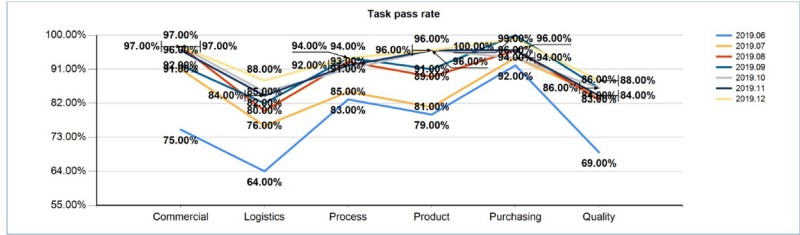

**Fig 9. Monthly post functional task closing rate (2019.12).**

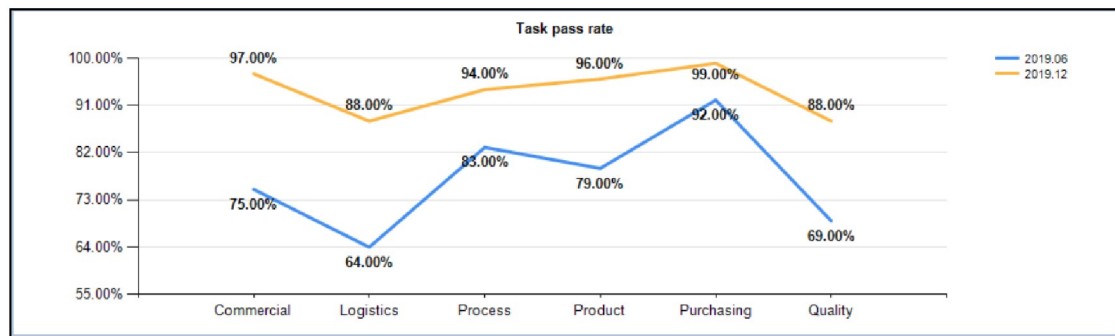

**Fig 10. Post functional task closure rate comparison (2019.6 VS 2019.12).**

daily habits after September 2019, and this trend has remained stable. Working in systems is becoming a daily habit for engineers. See Fig 9 for tracking data.

When the intermediate analysis process is removed, it can be seen that the shutdown rate of system tasks is significantly increased. See Fig 10 for comparison data.

**4.3.4 Task efficiency ratio change analysis.** With the increase in the task closure rate and the gradual formation of engineers' habits, it can be seen from the efficiency ratio of post functions and tasks that the completion habits of employees have greatly and steadily improved since September 2019. For tracking data, please refer to Fig 11.

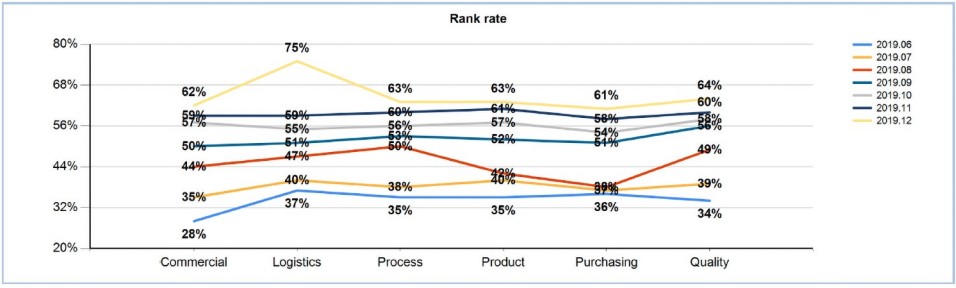

**Fig 11. Monthly post function task efficiency ratio (2019.12).**

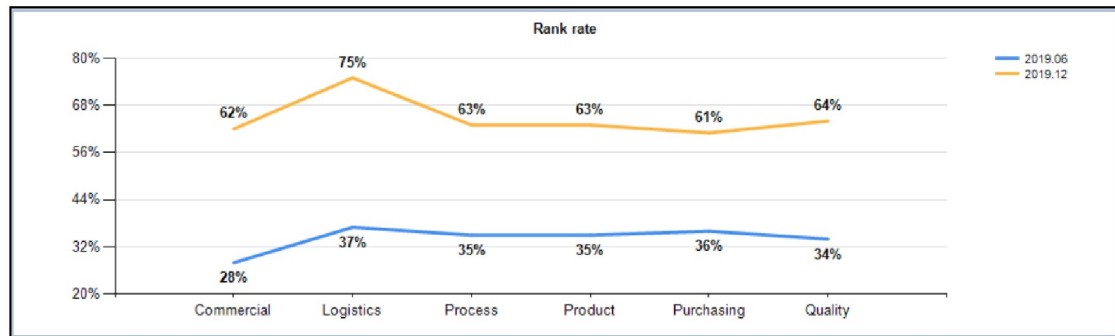

**Fig 12. Post functional task efficiency ratios (2019.6 VS 2019.12).**

By removing trace data per month, from June to December 2019, it can be seen that positions and functions have nearly 85% efficiency increases, the function of logistics tasks showed a 38% increase in efficiency, the function of the technology development of tasks showed a 28% increase in efficiency, the function of product development tasks showed a 28% increase in efficiency, the task of the purchasing function showed a 25% increase in efficiency, and the quality of functional tasks showed a 30% increase in efficiency, which are very surprising results. The comparison data are shown in Fig 12.

In addition to the significant improvement in the efficiency of job functions and tasks, we can see from the monthly tracking data that competition among teams is very fierce. The top three teams in terms of the on-time completion rates of tasks are almost the same every month, and the last three teams on the list are different. In such a process, competition among teams in the middle is particularly fierce; a user's activity directly promotes his/her habit of system operation, and the on-time rate of the task is also greatly increased. In terms of the on-time completion rate, the product development function increased by 37%, the process development function increased by 51%, the procurement function increased by 51%, the quality function increased by 35%, the business management function increased by 47%, and the logistics function increased by 48%. Refer to Fig 13 for comparative data.

**4.3.5 Analysis summary.** According to the content of the data analysis process and combined with the above data output, the following conclusions can be drawn:

(1) Analysis of employee performance level: the proportion of A-level employees increased by 18%, that of B-level employees increased by 21%, that of C-level employees increased by

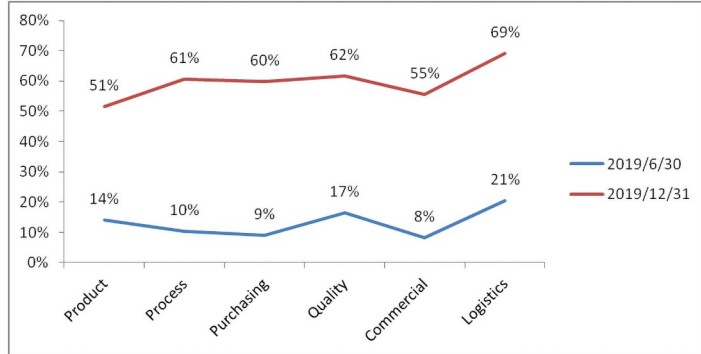

**Fig 13. Task on-time completion rate (2019.6 VS 2019.12).**

23%, and that of D-level employees decreased by 62%, indicating that the model achieved its initial target.

(2) Proportion distribution of employees' levels: the proportion of employees in development product level 5 increased by 14%, that in the development process increased by 11%, and that in business management increased by 11%. It can be concluded that the enthusiasm of engineers in completing tasks in the system is significantly improved. Moreover, this result indicates that the performance evaluation model of this study has high usability.

(3) Change analysis of the task closure rate: after half a year of whole-process tracking analysis, the task closure rate increased from 73% at the beginning to 96% at the end of the year, indicating that the performance evaluation model plays a role in promoting task closure.

(4) Task efficiency change analysis: from June to December 2019, the efficiency of positions and functions increased nearly 85%, that of the function of logistics tasks increased by 38%, that of the function of process development tasks increased by 28%, that of the function of product development tasks increased by 28%, that of the purchasing function tasks increased by 25%, and that of the function of quality tasks increased by 30%, indicating that the implementation of the performance evaluation model effectively promoted the self-awareness of employees.

## 5. Conclusions

According to the tracking situation over the 6 months studied, this research has achieved the expected effect and solved the three main problems in the process of project completion.

(1) Habitual problems in terms of employees' use of performance evaluation systems. After three months of management requirements and the learning, acceptance and formation of daily working habits, the project team is in line with the internalized transformation process of self-cognition theory and can give priority to the task operation of the system in the process of its daily use. For project management, this greatly reduces the coordination difficulty in the project management process. In the process of daily use, the project manager mainly focuses on the project time and tracks the key tasks of the project in a timely manner to confirm the specific situation of the project and focuses on real project time management. Focusing on energy also makes project time planning very reasonable.

(2) The implementation of process planning and methods. Throughout the course of the study, team members are listed as all being subordinate to leadership. The participation of functional leaders greatly promotes the system use and enthusiasm of project team members, the function of leadership is also easier to visualize due to the process of managing a weak point. For staff members, they check their work tasks and directly delay such tasks. Moreover, the timely communication of the staff also promotes the system management process.

(3) Post functions and project promotion. In the process of its application, this model solves the problem of task output quality in the project process and the problem of function division of each post within the project. The task efficiency ratio was 55% of the original goal and finally reached an average of 65%, indicating that the performance model and ranking had a positive and significant impact on the motivation of employees. Moreover, through this study, users also changed from passive to active users, which reduced management pressure and better reflected employees' strengths.

PLM systems have the following advantages in daily project management:

(1) It is conducive to the formation and transformation of users' habits.

(2) More fair and transparent assessment of employees' performance and ability.

(3) Provide a reference scheme and basis for enterprise management to select the best and eliminate the best.

(4) The system is aimed at enterprises that have entered a stable growth period, and can solve the "lazy" phenomenon of veteran employees.

Digital and networked management has become the mainstream trend of modern enterprise management, in which the data that can be analyzed and assessed are increasingly enriched, as is the assessment content of employees. Although this study has solved the problem of employees' habits of using the system, with the gradual deepening of the use of the system, the assessment parameters are not enough to support employees' dependence on and recognition of the system assessment. Therefore, in a follow up study, more factors will be added as the weighted data of performance analysis, including customer factors, team capabilities, market environment, etc., and an integrated assessment will be carried out from the multifunctional dimension to ensure the fairness and objectivity of the performance assessment.

## Supporting information

**S1 Data.**
(XLSX)

## Author Contributions

**Conceptualization:** Yinxia Ma.

**Resources:** Shihao Wang.

**Software:** Shihao Wang.

**Supervision:** Debao Dai, Yinxia Ma.

**Validation:** Yinxia Ma.

**Writing – original draft:** Shihao Wang.

**Writing – review & editing:** Debao Dai.

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
