## [Decision Letter · Decision Letter 0]

18 May 2021

PONE-D-21-01779

Performance evaluation of automotive product development team members based on PLM system: A case study of M Automotive Products Company

PLOS ONE

Dear Dr. yinxia,

Thank you for submitting your manuscript to PLOS ONE. After careful consideration, we feel that it has merit but does not fully meet PLOS ONE’s publication criteria as it currently stands. Therefore, we invite you to submit a revised version of the manuscript that addresses the points raised during the review process.

We look forward to receiving your revised manuscript.

Kind regards,

Anandakumar Haldorai, PhD

Academic Editor

PLOS ONE

Journal Requirements:

4. Please amend your authorship list in your manuscript file to include author Ma yinxia.

Additional Editor Comments:

Following are the reviews for your paper referenced above. I invite you to revise and resubmit your paper within 30 days (independent of what the system shows). Please carefully address the issues raised in the comments and, up front in your revised paper, describe how the comments of each reviewer are addressed. Your revised paper will be sent to the same reviewers, as well as possibly new reviewers, for evaluation.

You may ignore any suggestion of including self-references by reviewers if not applicable.

It is recommended to use a professional native English-speaking editor. Papers with less than excellent English will not be published even if technically perfect.

Include a paragraph at the end of the Introduction describing the organization of the paper.

Make sure the Abstract briefly describes the paper as it is used in abstracting and citation services. Keep the Abstract between 200 words. Do not use any references in the Abstract.

Spell out each acronym the first time used in the body of the paper. Spell out acronyms in the Abstract only if used there.

Include a list of six to ten key words after the Abstract.

Reviewers' comments:

Reviewer's Responses to Questions

**Comments to the Author**

1. Is the manuscript technically sound, and do the data support the conclusions?

Reviewer #1: Yes

Reviewer #2: Yes

2. Has the statistical analysis been performed appropriately and rigorously? 

Reviewer #1: Yes

Reviewer #2: Yes

3. Have the authors made all data underlying the findings in their manuscript fully available?

Reviewer #1: Yes

Reviewer #2: Yes

4. Is the manuscript presented in an intelligible fashion and written in standard English?

Reviewer #1: Yes

Reviewer #2: Yes

5. Review Comments to the Author

Reviewer #1: The manuscript is interesting. The language is engaging. The structure is good.

The authors are suggested to better highlight the contribution of the study. .

Cite more recent similar articles.

Please make sure that providing a separate section for literature review conform to the journal format. If not please combine the literature review with the introduction.

It is recommended to compare the proposed model with other similar models to highlight more the advantage of the system.

highlight the limitation of the study.

Good luck

Reviewer #2: The paper is well written and novel, making a good contribution. To make the paper easier to comprehend I suggest the authors improve the introduction part and Result & discussion part . Also , add some citation in these parts

6. PLOS authors have the option to publish the peer review history of their article (what does this mean?). If published, this will include your full peer review and any attached files.

Reviewer #1: No

Reviewer #2: No

---

## [Author Response · Author response to Decision Letter 0]

28 Jun 2021

Response to Reviewers

Dear Editors and Reviewers:

Thank you for your letter and for the reviewers’ comments concerning our manuscript entitled “Performance evaluation of automotive product development team members based on PLM system: A case study of M Automotive Products Company” (ID: PONE-D-21-01779). Those comments are all valuable and very helpful for revising and improving our paper, as well as the important guiding significance to our researches. We have studied comments carefully and have made correction which we hope meet with approval. Revised portion are marked in red in the paper. The main corrections in the paper and the responds to the reviewer’s comments are as flowing:

Responds to the reviewer’s comments:

Reviewer #1:

1. Response to comment: Highlight the contribution of the study

Response: Considering the Reviewer’s suggestion, we focused on the research value of this paper from two aspects of social value and academic value in the 1. Introduction part. 

2. Response to comment: Highlight more the advantage of the system and the limitation of the study

Response: In 5. Conclusion, we summarize four advantages of PLM system and three disadvantages of this research.

3. Response to comment: Cite more recent similar articles

Response: In order to better demonstrate the superiority of the model and system and show more theoretical support, this paper quotes 18 new literatures related to project management, system development and performance evaluation.

Special thanks to you for your good comments.

Reviewer #2:

1. Response to comment: Improve the introduction part and Result & discussion part

Response: We have re-written this part according to the Reviewer’s suggestion. In the 1.Introduction part, we state the research methods and ideas of this paper in detail. In the 3. Method part, we elaborate the theory basis and model of PLM system development. In the 5. Conclusion part, we emphasize the advantages of PLM system and the research defects of this paper.

2. Response to comment: add some citation in introduction part and Result & discussion

Response: In order to better demonstrate the superiority of the model and system and show more theoretical support, this paper quotes 18 new literatures related to project management, system development and performance evaluation.

Special thanks to you for your good comments.

We tried our best to improve the manuscript and made some changes in the manuscript. These changes will not influence the content and framework of the paper. And here we did not list the changes but marked in red in revised paper.

We appreciate for Editors/Reviewers’ warm work earnestly, and hope that the correction will meet with approval.

Once again, thank you very much for your comments and suggestions.

---

## [Decision Letter · Decision Letter 1]

14 Jul 2021

Performance evaluation of automotive product development team members based on PLM system: A case study of M Automotive Products Company

PONE-D-21-01779R1

Dear Dr. yinxia,

We’re pleased to inform you that your manuscript has been judged scientifically suitable for publication and will be formally accepted for publication once it meets all outstanding technical requirements.

Kind regards,

Anandakumar Haldorai, PhD

Academic Editor

PLOS ONE

Additional Editor Comments (optional):

Recommended for further publication process.

Reviewers' comments:

Reviewer's Responses to Questions

**Comments to the Author**

1. If the authors have adequately addressed your comments raised in a previous round of review and you feel that this manuscript is now acceptable for publication, you may indicate that here to bypass the “Comments to the Author” section, enter your conflict of interest statement in the “Confidential to Editor” section, and submit your "Accept" recommendation.

Reviewer #1: All comments have been addressed

Reviewer #2: All comments have been addressed

2. Is the manuscript technically sound, and do the data support the conclusions?

Reviewer #1: Yes

Reviewer #2: Yes

3. Has the statistical analysis been performed appropriately and rigorously? 

Reviewer #1: Yes

Reviewer #2: Yes

4. Have the authors made all data underlying the findings in their manuscript fully available?

Reviewer #1: Yes

Reviewer #2: Yes

5. Is the manuscript presented in an intelligible fashion and written in standard English?

Reviewer #1: Yes

Reviewer #2: Yes

6. Review Comments to the Author

Reviewer #1: (No Response)

Reviewer #2: The authors have successfully addressed all my concerns in the revised manuscript. Hence I recommend the acceptance of this paper.

7. PLOS authors have the option to publish the peer review history of their article (what does this mean?). If published, this will include your full peer review and any attached files.

Reviewer #1: No

Reviewer #2: No

---

## [Editor Report · Acceptance letter]

22 Jul 2021

PONE-D-21-01779R1 

Performance evaluation of automotive product development team members based on a PLM system: A case study of M Automotive Products Company 

Dear Dr. Ma:

I'm pleased to inform you that your manuscript has been deemed suitable for publication in PLOS ONE. Congratulations! Your manuscript is now with our production department. 

Kind regards, 

on behalf of

Dr. Anandakumar Haldorai 

Academic Editor

PLOS ONE